# BROWSERARENA: EVALUATING LLM AGENTS ON REAL-WORLD WEB NAVIGATION TASKS

## ABSTRACT

LLM web agents now browse and take actions on the open web, yet current agent evaluations are constrained to sandboxed environments or artificial tasks. We introduce BrowserArena, a live open-web agent evaluation platform that collects user-submitted tasks, runs Arena-style head-to-head comparisons, and uses step-level human feedback to surface failure modes. Collecting and analyzing step-level annotations on the agent traces, we identify three consistent failure modes: captcha resolution, pop-up banner removal, and direct navigation to URLs. By constructing targeted datasets to further study these tasks, we discover variations in how different language models navigate these failure modes. We find, for example, that o4-mini deploys a wider variety of strategies to circumvent captcha resolution than other models and DeepSeek-R1 consistently misleads users about pop-up banner closure. Our findings surface both the diversity and brittleness of current web agents. More broadly, our benchmarking methodology provides an approach to evaluating and understanding web agent failure modes at scale.

## 1 INTRODUCTION

Recently, with the advent of web agents such as Manus and OpenAI's Operator (OpenAI, 2025), there has been significant interest in the ability of large language models (LLMs) to interact and complete tasks on diverse websites. As a result, several benchmarks have been developed to evaluate the performance of various LLMs and agent frameworks on web browsing tasks (Yehudai et al., 2025). Some of these benchmarks focus on agent interaction with self-hosted websites, with success on tasks being measured using custom execution-based evaluation procedures (Koh et al., 2024). However, "closed" benchmarks have limited task diversity (Yoran et al., 2024) because they are restricted to only a few websites, so current benchmarks cannot serve as good tests of real-world web agents.

**Limitations of current open-web evaluations:** Recently, researchers have built systems that allow agents to browse the open web (Chezelles et al., 2024; Wang et al., 2024), given the significant success of open-ended environments for agent evaluation in other domains such as software engineering (Wang et al., 2024) and general computer use (Bonatti et al., 2024; Xie et al., 2024). However, such approaches still suffer from four major drawbacks. First, in such benchmarks, tasks are described using highly specific instructions to the agent, which is unlikely to mirror how real-world users describe and perform tasks on the open web. Second, significant engineering effort is often required to incorporate new tasks into these systems because they often require ground-truth success criterion for measuring task performance (Chezelles et al., 2024). This need for ground-truth success criteria limits the types of tasks that can be evaluated within these approaches. Third, since these success criteria are often evaluated using programs, they also serve as an entry barrier preventing non-technical users from contributing new tasks to these benchmarks. Due to this entry barrier, most benchmarks developed on top of such open-web environments are static, ground-truth-based benchmarks with detailed task descriptions. Finally, existing ground-truth based benchmarks can be accessed by a diverse range of LLMs with different levels of tool access and reasoning frameworks as long as the final system produces the correct ground truth result. While this flexibility is helpful for comparing across a wide range of systems, it obscures the differences in performance due to the usage of different language models.

**Our approach: A live evaluation platform using user-submitted tasks and pairwise comparison between agents.** We introduce BrowserArena, a live evaluation platform for evaluating LLM

performance on user-submitted open-ended web agent tasks which builds off the Chatbot Arena (Chiang et al., 2024) framework. In BrowserArena, users are requested to enter a task description, which is then submitted to two randomly-selected LLMs that utilize the BrowserUse library (Müller and Žunič, 2024) to interact with and navigate different websites. BrowserArena uses a similar evaluation approach to other platforms for open-ended tasks, such as Chatbot Arena (Chi et al., 2025) and Copilot Arena (Chiang et al., 2024): pairwise comparisons between different agents to develop models for human preferences. This approach allows for the evaluation of tasks with ambiguous specifications and allows users to rank agent outputs according to criteria that may be difficult to evaluate in a ground-truth-based benchmark (such as whether the intermediate steps taken by the agent were reasonable).

**Can VLMs model human preferences on agent performance?** After collecting the user-submitted votes, we ask a new set of users to evaluate a subset of the original user-submitted tasks to measure the variance in user preference while evaluating the same agents on the same task. We observe that there is broad agreement with the original user-submitted preferences while taking a majority vote among the new users' submissions. However, despite previous work demonstrating multimodal-LLM-as-a-judge capabilities on evaluating pair comparisons on other image-based datasets (Chen et al., 2024), our experiments show that there is still a significant gap between human preferences and the preferences exhibited by vision-language models (VLMs).

**Identification of agent failure modes through user-submitted step-level feedback:** To overcome VLMs' limited capabilities for evaluating agents, we present an alternative methodology using human step-level feedback for identifying "failure modes" Brown et al. (2025); Meng et al. (2025), which are recurring situations across different tasks where users report that LLM agent behavior did not meet their expectations. Our approach is as follows: in our study, after a user submits a task on our evaluation platform, we ask the same user to annotate the steps produced in both agents' output traces to understand where the agent may have fallen short of user expectations. Intermediate steps in agent traces contain LLM-generated stepwise goals as well as descriptions of the actions taken during that step. We ask users to either mark the step's actions as correct with respect to its goal or mark it as incorrect and explain why it is contrary to their expectations of a successful step. This approach helps us collect more granular insights into agent behavior when compared to the simple voting mechanism present in prior work (Chiang et al., 2024). By analyzing user-submitted annotations, we identify three failure modes occurring within our system (captcha resolution, pop-up banner removal, and direct navigation to URLs). We then construct targeted datasets of tasks which reproduce these failure modes with a high frequency, and present our conclusions on the differences in language model behavior on these failure modes. We currently plan to open-source the BrowserArena platform codebase for collecting preference data to help identify new agent failure modes.

Our key contributions are as follows:

1. We present an evaluation platform, BrowserArena, for pairwise comparison between models for user-submitted web-browsing tasks (Section 3).

2. We collect user preference data on 109 user-submitted tasks, using which we construct a language model leaderboard and demonstrate a gap in existing VLMs' ability to model human preferences (Section 4).

3. Given VLM preference labeling unreliability, we describe a new methodology for evaluating language model performance in web browsing by collecting step-level user annotations on agent traces and analyzing them to identify common failure modes, which are then studied separately (Section 5). We find, for example, that DeepSeek-R1 consistently misrepresents its ability to close pop-up banners, despite being unable to even identify such banners (due to its lack of multimodal capabilities).

## 2 RELATED WORK

**Question Answering Benchmarks:** Several popular web agent benchmarks formulate their tasks as text or multimodal inputs to question-answering systems since they can be evaluated using reference ground truth strings. AssistantBench (Yoran et al., 2024) presents a dataset of user-submitted domain-specific text-only QA tasks which only accept strings, numbers, and dictionaries as ground truth. WebQA (Chang et al., 2022) comprises of multi-image and complex single-image questions presented

to the model alongside a set of positive sources and distractor sources. GAIA (Mialon et al., 2023) presents a QA benchmark with more difficult tasks, several of which either require web browsing, code execution, and diverse filetype reading capabilities. BrowseComp (Wei et al., 2025) comprises of even harder QA tasks which take humans several hours of browsing to solve since the correct answers to the questions satisfy several constraints that are difficult to evaluate. While these benchmarks can evaluate agents' ability to search the web for information that may be very difficult to find or reason about data discovered via web search, they do not accurately represent how most human users would use these models for web navigation tasks on an everyday basis and does not measure several abilities valued by humans while navigating the web, such as navigating and taking actions on dynamic websites.

**Self-Hosted and Simulated Benchmarks:** Mind2Web (Deng et al., 2023) uses real-world webpage snapshots that include raw HTML code, DOM snapshots, and the network traffic for replaying an interaction, but formulates the web navigation task as an action selection or element selection task, restricting their measure of success to successfully replicating human-generated trajectories. Other approaches have formulated the web navigation problem as Partially-Observable Markov Decision Processes (POMDPs) with various reward mechanisms. For example, WebShop (Yao et al., 2022) introduced a simulated environment for executing search tasks defined in natural language on a shopping website containing products listed on Amazon, with agents only allowed to take click and search actions with ground truth rewards based on product attributes. WebArena (Zhou et al., 2023) introduced a benchmark for executing natural language tasks on four self-hosted clones of popular websites with a larger action set, using both ground-truth answers and LLM-guided fuzzy matching for evaluating agent success. WebArena has been extended for evaluating agents on visually-grounded tasks in VisualWebArena (Koh et al., 2024), on tasks involving learning from long-context video understanding in VideoWebArena (Jang et al., 2024), and on complex tasks requiring mathematical reasoning and memory in WebChoreArena (Miyai et al., 2025) using similar evaluation procedures. However, these benchmarks assign rewards based on the final output produced by the trajectory, making it difficult to assess if the intermediate steps taken by the agent would be considered reasonable by humans (as they assign equal rewards to two agents even if they take different approaches to reaching the same terminal state). They also do not provide methods for evaluating partial progress on tasks that are grounded in human preferences, instead relying on fuzzy matching to reward models whose outputs resemble the ground truth at the end of their trajectory.

**Open Web Benchmarks:** Certain popular benchmarks have adapted their evaluation methodology for evaluating web agents that can browse on the open web. WebVoyager (He et al., 2024) introduces a benchmark comprising tasks from 15 websites, omitting websites requiring CAPTCHA or login, developing tasks by sampling and rewriting tasks from Mind2Web (Deng et al., 2023) and prompting LLMs to generate new tasks. However, they then have to annotate tasks with sets of possible answers, with only 22.3% of tasks having "golden" answers that they expect not to change in the short term. MMInA (Zhang et al., 2024) converts tasks from the WebQA dataset (Chang et al., 2022) into multimodal multi-hop problems, annotating them with instructions, examples of other QA tasks, and a "universe" of websites the model is allowed to visit while solving the tasks. While single-hop tasks are evaluated using ground-truth and fuzzy-matching based evaluations, multi-hop tasks are evaluated by marking tasks as completed only if each hop was completed correctly (by either visiting the correct link or by collecting the desired information). Thus, despite evaluating on dynamic, changing websites, the benchmark is restricted to evaluating tasks with respect to either ground-truth information or human-defined trajectories, making it difficult to scale the benchmark construction methodology to new tasks and websites (especially given the benchmark's dependence on dynamic web content not breaking the ground-truth human trajectories). SearchArena (Miroyan et al., 2025) is an extension of ChatbotArena's user-preference guided leaderboard system that allows for users to evaluate tasks on two randomly selected LLMs augmented with search capabilities. However, their framework is restricted to web search tasks for retrieving and summarizing information, and is unable to evaluate web agent behavior on browser-based tasks involving taking actions on websites. Additionally, SearchArena does not provide agent traces describing the sequence of websites visited and actions taken on each website, making it difficult to compare partial progress on each task and analyze step-level feedback.

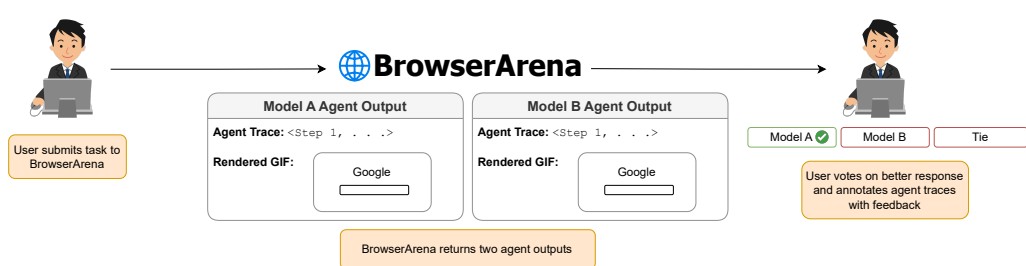

Figure 1: An overview of the study procedure showing how users interact with BrowserArena. We include examples of user submitted tasks in Appendix M.

## 3 BROWSERARENA EVALUATION PLATFORM

We develop the BrowserArena website by equipping ChatBot Arena's open-source codebase (Chiang et al., 2024) with the capabilty of submitting a task to BrowserUse (Müller and Žunič, 2024) and visualizing the results. On visiting the website, users are presented with a text box in which to enter a description of the task (examples of user-submitted tasks are in Appendix M). Once the user submits their task, two LLMs are chosen at random with uniform probability for creating the BrowserUse agents. These models are then used to construct BrowserUse agents, which utilize independent Playwright (Microsoft, 2025) instances for automating a Chromium browser. The LLM is permitted to choose an action from the set of actions pre-defined by the BrowserUse controller (for a full list, see Table 1 in Appendix C). The BrowserUse agents accept the task, previous steps, current URL, open tabs, and a list of HTML elements with associated numeric indices, where the indices of interactive elements are distinguished from the other elements. If the model has multimodal capabilities (all our tested models except DeepSeek-R1), it also receives a screenshot of the current browser with an overlay labelling the rendered HTML elements with their indices. The LLMs then output a JSON object describing the current state of the task, containing four properties: a self-evaluation of whether the previous goal was completed, a memory property describing what has been done so far, a goal property describing the next immediate objective, and a sequence of actions to take.

The user-submitted task prompt is then submitted to the two BrowserUse agents, each using one of the sampled LLMs as the model backend. Once both models finish, we present the user with the agent outputs of the models, as well as a GIF rendering each step that the agent took on the Playwright Chromium browser instance. Once these agent outputs are rendered on the website, users are provided with an option to vote on which response is better.

## 4 EXPERIMENTAL EVALUATION

For collecting tasks on BrowserArena, we first design a user study (details described in Section 4.1) asking users to submit tasks, vote for the agent that best completed the task, and annotate the generated agent traces. Then, using user votes, we construct a leaderboard of models . We present our results in Section 4.2. Then, we run a study to measure human evaluator agreement on a subset of the user-submitted tasks (detailed in Section 4.3), and demonstrate a significant gap between VLM preferences and human preferences based on agent outputs in Section 4.4.

### 4.1 USER STUDY DESIGN

For our experimental study, we solicit tasks and feedback on agent performance via a survey on Prolific. We recruit users on Prolific from United Kingdom, United States, Australia, Canada, and New Zealand with response approval rates between 90–100%. We approved a total of 213 valid responses, ultimately keeping 109 responses from 98 users due to system outages, logging issues, and invalid responses. We collected responses in 3 batches, with the average of the batch median completion times being 35:10 minutes, and payments being made at an average hourly rate of $8.01/hr.

We provide further details about the each batch's compensation and median completion times in Appendix H.

We ask Prolific users to submit tasks that involve clicking and interacting with different websites (which we call "interactive tasks") and to explicitly avoid submitting tasks which either can be answered by analyzing Google search result links and descriptions or can be answered by a language-model chatbot without searching for an answer. We also caution users not to enter tasks where the answer is easily provided within the Google search results without clicking on a website or is an open-ended question that a chatbot can answer without clicking on any website (which we call "search tasks"). We provide some examples to help users differentiate between the two, which we have listed in Appendix A.

Since the goal of each step is LLM-defined, we ask users to use the agent traces and the generated GIFs to identify steps that were executed correctly with respect to their goals, and describe where "incorrect" steps fell short. With the help of user feedback, we analyze the agent traces and construct a mapping between each step generated by the agent and whether it was perceived to be successfully executed. This information is then used to identify the failure modes explored in our case study in Section 5. Finally, users are asked to vote between the two LLM models. Unlike the original ChatBot Arena website, we only accept "Left", "Right", and "Tie" votes and ignore "Both models are bad" votes, since we are interested in measuring partial progress if both agents fail.

We then utilize the user votes to construct a model leaderboard of voter preferences. We evaluate the performance of five models on the BrowserArena platform: DeepSeek **R1**, Anthropic **Claude 3.7** Sonnet:Thinking, Meta **Llama-4**-Maverick, OpenAI **o4-mini**, and Google **Gemini 2.5**-Pro-Preview-03-25 using the OpenRouter API platform. In our subsequent discussions, we will refer to each of these models by the bolded portion of their names. We note that while the BrowserUse platform supports submitting image screenshots of the webpage alongside search results and web page structure in API calls to the model, R1, being a language model without multimodal capabilities does not utilize the image screenshot provided.

## 4.2 Ranking Results

By estimating the Bradley-Terry coefficients of each model (Bradley and Terry, 1952) based on the user votes, we compute the ranks of different models using the ranking methodology described in Chatbot Arena (Chiang et al., 2024). We provide a more detailed summary of leaderboard construction in Appendix B. We present our leaderboard from 109 valid battles alongside our win fraction heatmap, average win rate bar, confidence interval calculations, and a heatmap of the battlecounts in Figure 2. Based on the user-submitted tasks, the LLM agent with the highest ELO rating is based on R1, which surprisingly is the only model evaluated that does not have multimodal capabilities.

## 4.3 Human Evaluator Agreement

We evaluate how consistently humans judge head-to-head browser-agent runs on 25 randomly selected task submissions, and find modest-to-strong agreement. For each task, annotators are shown the same agent trace and GIF comparison used for the original task submissions, and are asked to select between Agent 1, Agent 2, and Tie. 165 new human annotations are collected from Prolific; we use two screening questions and participants take on-average 57 seconds to provide a selection on a task. We compare these human aggregates to the label from the original task submission with inter-annotator agreement, which measures how often different human evaluators make the same choice when comparing two agent trajectories, with higher agreement indicating clearer differences in performance between the agents. We find that the majority vote of the new human annotators has modest agreement with the baseline labels (63.2% of questions) and modest inter-annotator agreement (57.6%). Lower agreement is largely explained by the lack of consistency between labelers when voting 'tie'; the majority vote agreement goes up to 100% agreement when 'tie' votes are removed and we force a majority selection between Agent 1 and 2. Similarly, the inter-annotator agreement goes up to 83% when ties are filtered. These results suggest that differences between human agent judgements reflect differing decision thresholds more than differing rank orderings of agents.

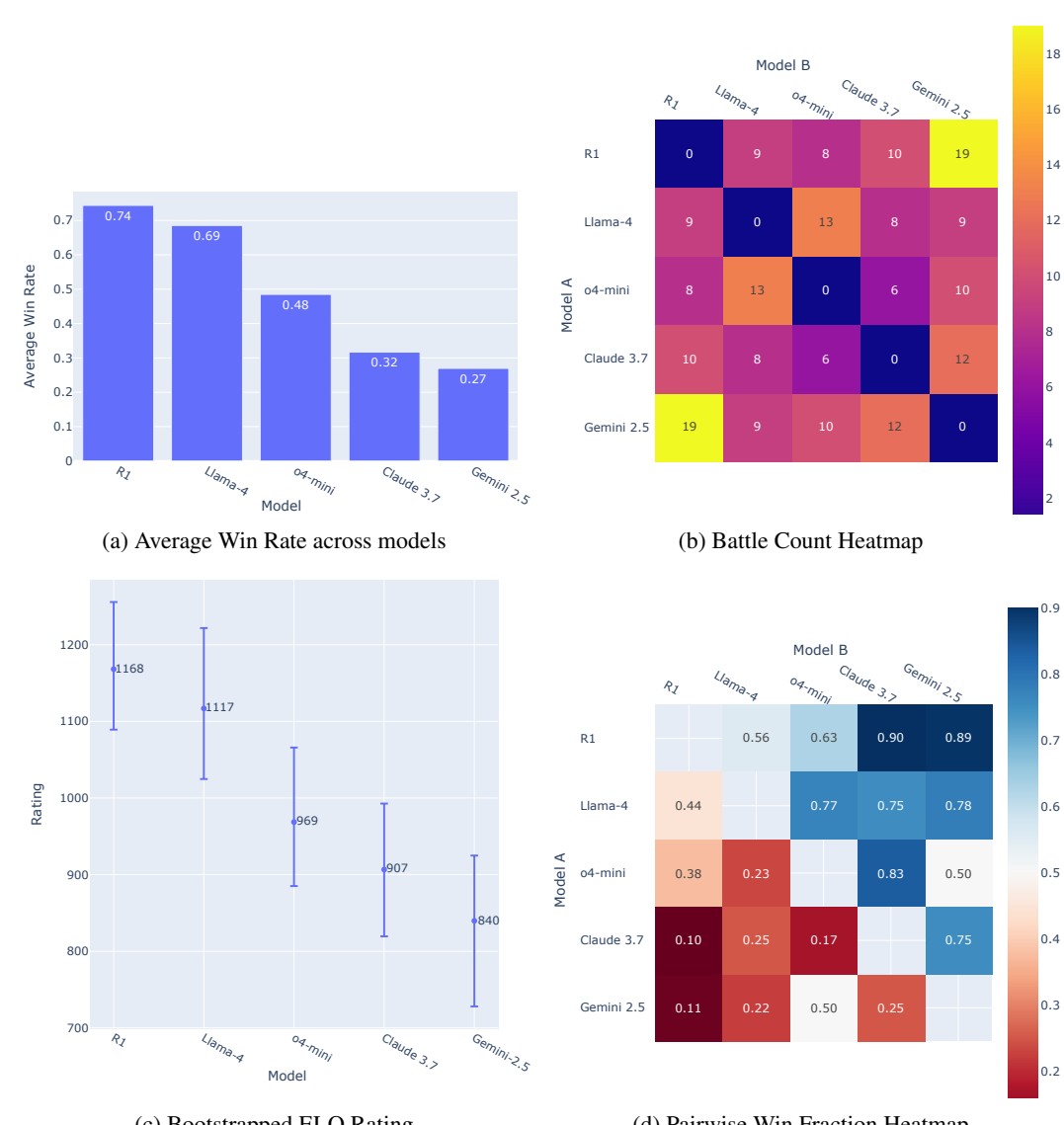

(a) Average Win Rate across models

(b) Battle Count Heatmap

(c) Bootstrapped ELO Rating

(d) Pairwise Win Fraction Heatmap

Figure 2: We compute the average win rate, battle counts, bootstrapped ELO ratings, and pairwise win fractions from 109 user-submitted tasks and evaluations on Prolific. For the ELO-based leaderboard, we simply sort the models from highest to lowest bootstrapped ELO rating in Figure 2(c).

## 4.4 VLM-AS-A-JUDGE

For VLM evaluation, we use the same 25 randomly selected task submissions we use for measuring human evaluator agreement. The original human task labels are compared to two vision-language model judges (GPT-4o, o4-mini) that are prompted with the same input (the agent trace and GIFs) and asked to choose between select between Agent 1, Agent 2, and Tie. As shown in Figure 3, GPT-4o has relatively high agreement with the human annotation baseline (68%), o4-mini only 58%. Interestingly, we find that the GIFs showing the agent computer seem to be hurt GPT-4o agreement: in input ablations, trace-only evaluation improves GPT-4o's agreement with the baseline annotations by 10 percentage points (79% vs. 68% with GIFs and traces), while GIF-only input collapses performance to 48% agreement despite an increased self-reported confidence. These results indicate that multimodality can hurt judge reliability in this setting. In summary, we find a sizable gap in labeler agreement between VLMs and humans.

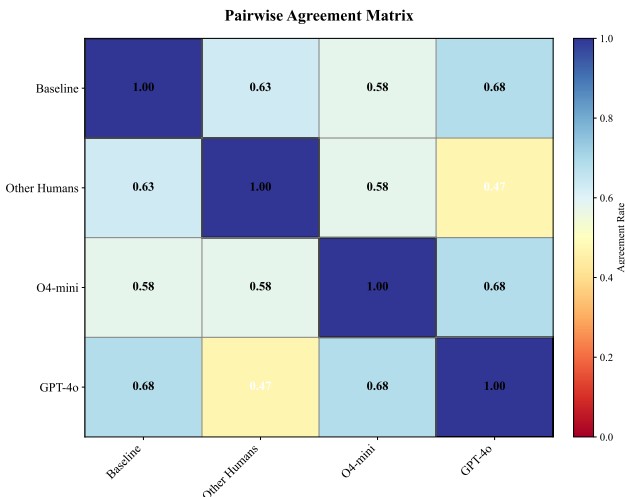

Figure 3: Pairwise agreements between the baseline labels, the new annotators, and two vision-languages models (GPT-4o and o4-mini; we take the majority @5).

## 5 PROMINENT AGENT FAILURE MODES

We use the agent traces and human feedback collected in our benchmark to surface and study three prominent failure modes in current agents. After collecting the step-level feedback as a part of our initial Prolific user study, we cluster and summarize the step-level annotations as described in Section 5.1. Using these clusters, we identify three failure modes where agents fail to complete tasks which we investigate in greater detail: Captcha Solving (Section 5.2), Pop-Up Banner Closure (Section 5.3), and Direct Navigation (Section 5.4).

To study variations in model behavior on occurrence of each of these failure modes, we use the following general pipeline. We first construct a new larger dataset of tasks which reproduce the failure mode scenario with a high probability when an agent attempts to complete the task. Then, once we execute these tasks for each language model, we use o4-mini as a judge to evaluate the traces generated by the agent and determine if the specific failure mode occurred while the agent was executing the task. We then report aggregate statistics on how often each language model ran into specific scenarios while executing the tasks.

### 5.1 DISCOVERING COMMON FAILURE MODES

We use our step-level human labels on the BrowserArena agent tasks to automatically find 'failure modes' (Meng et al., 2025; Brown et al., 2025), consistent mistakes an agent makes while performing the user-submitted tasks. Three of our discovered failure modes are explored in detail in Sections 5.2-5.4. To automatically find these common failure modes, we use two methods (dataset featurization (Bravansky et al., 2025) and an API-only method, Docent (Meng et al., 2025)) that first cluster the step-level labels in an embedding feature space, and then use auxiliary LLMs to summarize these clusters; these cluster summaries, which pick out consistent agent behavior across tasks, are the failure modes. The methods find very similar failure modes; we give the full set of discovered failure modes found via dataset featurization in Table 2 and Docent in Figure 4(a). Our cluster and summarization hyperparameters are described in Appendix I.

We then select the following three failure modes for a more detailed investigation from the list of failure modes we have constructed:

1. **Captcha Solving:** On encountering a captcha puzzle, agents can get stuck while attempting to solve the puzzle since the individual components of the puzzle may not be clickable elements in the webpage's DOM. We thus seek to study the different strategies used by different language models to evaluate if specific models prefer different captcha-avoidance methods to others.

2. **Pop-Up Banner Closure:** On encountering a pop-up banner obscuring a part of the website, agents can be preventing from making progress on the remainder of the task due to being unable to close the pop-up-banner. We thus study how often a language model identifies that a pop-up banner is blocking its access to the website and successfully closes the banner and moves ahead with its task.

3. **Direct Navigation**: Sometimes, agents choose to directly navigate to a website URL (hereby referred to as the starting website) that they believe is integral for solving the task as opposed to conducting a Google Search to collect relevant links first. This can lead to delays in completing the task if navigating the starting website is more complex for the agent compared to the websites which may have been selected had the model conducted a Google search first.

## 5.2 Captcha Solving

**Dataset Construction:** We first identify `www.expedia.com` as a website that is reliably blocked by a captcha on our system when an agent attempts to visit it while solving a user-submitted task. We then construct a dataset of 220 tasks which require interacting with or visiting the Expedia website. 20 of these tasks are constructed from human written task templates, and 200 of them are generated by GPT 4.1 using a task generation prompt (for template and prompt details, see Appendix D).

**Scenarios:** We first construct a set of captcha circumvention strategies by manually examining LLM agent traces produced by different models on some of the 20 template-based tasks. We additionally have an LLM (o4-mini) also analyze all of these traces and identify if any other strategies have been used for captcha navigation in these traces. We add the new strategies detected by LLM to our existing set of strategies. Finally, we use o4-mini to identify if any strategy from our strategy set was used in the agent traces of each of the 220 tasks that each LLM attempted to solve. For the detailed prompt used for o4-mini to judge all the agent traces and the description of each strategy provided in the prompt, see Appendix F.

**Results:** We present our results measuring the percentage of times each particular strategy was deployed by a model in Table 3 in Appendix J. We observe that most language models show a clear preference for the "Direct Link", "Google Search", and "New Tab" strategies. However, Claude 3.7 prefers the Switch Websites method much more than other LLMs, while both it and Gemini-2.5-Pro use the "New Tab" tactic less than other LLMs (and in fact prefer the "Switch Websites" method to it). On the other hand, o4-mini uses all the listed strategies at least once, and uses some strategies not used at all by other language models, such as the "Text-only Rendering", "Public Proxy", and "Internet Archive". It also uses tactics such as "Cache", "Mobile", and "Internal Navigation" and "Country Domain" at much higher rates than other LLMs, suggesting that it is better at getting around captcha challenges in the event of their presence disrupting the search than other language models as it is able to try a wider range of strategies.

## 5.3 Pop-up Banner Closure

**Dataset Construction:** We first identify `www.bbc.com` as a website that reliably generates a privacy policy banner when an agent attempts to visit it while solving a user-submitted task. We construct a dataset of 80 tasks which require interacting with or visiting the BBC website by prompting GPT 4.1 using a task generation prompt (for template and prompt details, see Appendix E).

**Scenarios:** We consider three scenarios that the LLM agent may find itself in: either it did not detect a pop-up banner while evaluating the task, it did discover a pop-up banner and successfully closed it, and it marked the task as being completed (independent of whether it managed to progress past the pop-up banner). We then use o4-mini to identify if any of these scenarios occurred in the agent traces of each of the 80 tasks that each LLM attempted to solve. For the detailed prompt used for o4-mini to judge all the agent traces and the description of each strategy provided in the prompt, see Appendix G. For complete results of how frequently each scenario occurs for specific LLMs, please refer to Table 4 in Appendix K.

**Results:** Notably, R1 seems to have never realized that a part of the website is blocked by a privacy policy pop-up in all the times it attempts to complete the BBC agent tasks, indicating that multi-modal reasoning ability is required for detecting the privacy policy pop-up. However, R1 marks the task as

completed at the highest rate of all the LLMs, suggesting that without multimodal capabilities, it is unable to reason that its task remains incomplete without closing the cookie banner. On the other hand, o4-mini and Llama-4 manage to close pop-up banners at a higher rate than the remaining multimodal LLMs, although only o4-mini marks a similar percentage of tasks as completed as compared to the percentage of tasks for which the LLM judge determines that it closed the pop-up banner.

### 5.4 DIRECT NAVIGATION

**Dataset Construction:** We focus on a knowledge-intensive question answering task to investigate whether agents opt to directly answer, directly navigate to relevant websites, such as Wikipedia, or instead invoke the Google Search API. To this end, we sample 100 questions from the TriviaQA dataset (Joshi et al., 2017), which comprises naturally occurring questions posed by trivia enthusiasts.

**Scenarios:** We consider two distinct scenarios that the language model (LLM) agents may encounter: (1) the agent recognizes the question and directly answers or navigates to the corresponding Wikipedia page; or (2) the agent lacks sufficient knowledge and first queries the web using Google Search. For each question, we collect the agent's execution trajectory and manually annotate the scenario it conforms to. A summary of the distribution of scenarios across models is provided in Table 5 in Appendix L.

**Results:** We observe that the most frequent behavior involves invoking the Google Search API to retrieve relevant information using extracted keywords. In some instances—more commonly observed with Llama-4—the agent navigates to Google.com and inputs search queries manually, rather than using the API. In contrast, direct answering or navigation to Wikipedia pages is relatively rare. These findings suggest that, in general, agents tend to follow the instruction and leverage Google as the primary information source when responding to knowledge-intensive queries.

## 6 CONCLUSIONS

In this paper, we have presented a web agent evaluation platform, BrowserArena, for pairwise comparison between various language models on user-submitted web browsing tasks. After collecting user preference data on 109 user-submitted tasks, we first construct a language model leaderboard to demonstrate user preferences between various models. Then, we demonstrate a gap between VLM agreement and human evaluator agreement on user preferences.

This gap motivates our development of a new methodology for evaluating language model performance by collecting step-level user annotations on agent traces and analyzing them to identify common failure modes. We then provide methods to construct three targeted datasets to further study these failure modes, and report our results on differences in model behavior when encountering these failure modes.

## 7 LIMITATIONS

Our approach for standardizing language model agents involves equipping models with BrowserUse (Müller and Žunič, 2024), which provides all models with a standard format in which to output their goals and the action to be taken in each step. However, equipping models with different or more powerful capabilities may help improve agent capabilities in solving tasks, which makes our results and evaluation method dependent on the browser agent system connected to the LLM.

Additionally, another drawback is that the failure modes we discover may be system specific. We believe that it is still useful to identify failure modes and construct targeted datasets to analyze model behavior under similar circumstances. However, the specific tasks that trigger the failure mode may be different depending on the system configuration - for example, it may be possible to reduce the likelihood of encountering captchas on a particular website by using rotating proxies.

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

## A EXAMPLE TASKS PRESENTED TO USERS

**Examples of Valid Interactive Tasks**:

1. What are today's top 20 headlines from CNN?
2. Compare the bus prices for one-way tickets from Boston to New York next Saturday on different ticket purchasing websites.
3. Create a list of the top-ranked chess players on chess.com from Belgium.

**Examples of Invalid Search Tasks** *(Alongside Why They are Invalid)*

1. How do I increase my concentration while working? *(This is invalid because it can be answered using a chatbot and does not require clicking on a specific website.)*
2. What is the weather today? *(Google will output this answer in a box displayed at the top of search results, again does not require clicking on a specific website.)*
3. Who are the members of the Beatles? *(Google provides a lot of links with the text containing the answer to this question under those links, so you do not need to click on a website to answer this question.)*

## B RANKING METHODOLOGY

We use a similar approach to other pairwise-comparison evaluation procedures for ranking models. Here, we present an overview of the procedure in the binary preference case for $M$ models. As defined in (Chi et al., 2025; Chiang et al., 2024), in a sequential setting, at time $t \in \mathbb{N}$, we first formally define our comparative data set $\mathcal{A} = \{(m, m') : m < m' \text{ and } m, m' \in [M]\}$. Then, for a pair of models $A_t = (i, j) \in \mathcal{A}$, we model the human preference $H_t \in \{0, 1\}$, where $H_t$ is 1 if $i$ is preferred over $j$ and 0 if $j$ is preferred over $i$. We then define the score function to be the vector of Bradley-Terry coefficients $\beta \in \mathbb{R}^M$ (Bradley and Terry, 1952). Under the Bradley-Terry model, the probability of model $i$ beating model $j$ i.e. $\mathbb{P}(H_t = 1)$ is given as shown:

$$\mathbb{P}(H_t = 1) = \frac{e^{\beta_i}}{e^{\beta_i} + e^{\beta_j}} \tag{1}$$

The rank of a model $m$ is then calculated as follows:

$$\text{rank}(\beta)_m = 1 + \sum_{m' \in [M]} \mathbb{1}\{\beta_{m'} > \beta_m\} \tag{2}$$

The BT coefficients are then estimated via maximum likelihood estimation, with 95% confidence intervals being calculated by bootstrapping for 100 rounds. After determining the confidence interval, the rank of each model is estimated by computing the number of models whose lower bound is less than its upper bound (Chi et al., 2025). This model can then be extended to cases when $H_t$ is not binary by estimating the BT score from a nonparametric extension of the Bradley-Terry model (Chiang et al., 2024).

# C  BROWSERUSE PERMITTED ACTIONS

| Action Name | Action Description |
|---|---|
| Complete Task | Mark task as completed with success=True if successfully completed and success=False if at last step. |
| Search Google | Search the query in Google on the current tab. |
| Go to URL | Visit the specified URL in the current tab. |
| Go Back | Go back in history to the previous website visited. |
| Wait | Wait for $x$ seconds where $x = 3$ by default |
| Wait for element to be visible | Wait for an element specified by the CSS Selector to become visible within the specified timeout. |
| Click element by Index | Click the HTML element specified by its numeric index |
| Click element by Selector | Click the HTML element specified by its CSS Selector. |
| Click element by XPath | Click the HTML element specified by its XPath path expression. |
| Click element with Text | Click the HTML element containing the provided text. |
| Input Text | Input the provided text into the specified input interactive element. |
| Save as PDF | Save the current page as a PDF file. |
| Switch Tab | Switch to a different browser tab. |
| Open URL in New Tab | Open the specified URL in a new tab. |
| Close Tab | Close the specified browser tab. |
| Extract Page Content | Extract page content using an LLM prompted with the specified goal. |
| Save as HTML | Save the raw HTML content of current page as an HTML file. |
| Scroll Down | Scroll down by a specified pixel amount, by default scroll down one page. |
| Scroll Up | Scroll up by a specified pixel amount, by default scroll up one page. |
| Send Special Keys | Send special key commands (Esc, Backspace, keyboard shortcuts) to the current page. |
| Scroll to Text | Scroll until the specified text is visible on the current page. |
| Get Dropdown Options | Get all options from a dropdown element. |
| Select Dropdown Option by Text | Select dropdown option using the specified text |
| Drag and Drop | Drag and drop elements, or between coordinates of the page. |

Table 1: BrowserUse actions available to LLMs, as specified in (Müller and Žunič, 2024).

# D  PROMPT USED FOR LLM-GENERATED CAPTCHA SOLVING TASKS

Note: While we prompt the language model to generate a 1000 tasks in most of our prompts, we usually get between 80-100 tasks in a single response. For generating our final list tasks, we prompt the model repeatedly and remove any duplicate tasks generated.

**Model Used:** GPT 4.1

**System Prompt:** You are a helpful task-generating agent.

**User Prompt:** In JSON format, generate a list of 1000 tasks for a web agent that involve navigating to the Expedia website and searching for information. Here are some examples of the desired tasks:

1. "Find a list of activities on Expedia to do in Chicago on December 3."

2. "Find the cheapest hotel in Houston on Expedia from August 8-12."

3. "Find the cheapest round-trip flights between Houston and Los Angeles on Expedia from November 10-15."

Make sure to randomize the cities, dates, and types of tasks. Do not generate any tasks that require the agent to take any action on the website except for searching and retrieving information. Make sure the tasks include the phrase "on Expedia" so that the agent searches for the Expedia website.

# E  PROMPT USED FOR LLM-GENERATED POP-UP BANNER TASKS

**Model Used:** GPT 4.1

**System Prompt:** You are a helpful task-generating agent.

**User Prompt:** In JSON format, generate a list of 1000 tasks for a web agent that involve navigating to the bbc.com website and searching for information. Here are some examples of the desired tasks:

1. "Find the top 10 headlines of the day on bbc.com."

2. "What is the top story today in the Culture section on bbc.com."

3. "List all the headlines from three days ago in the innovation section on bbc.com."

Make sure to randomize the number of days and the sections of the bbc.com website in the task. You must only choose sections from the following list: News, Sport, Business, Innovation, Culture, Arts, Travel, Earth, Audio, Video. Do not generate any tasks that require the agent to take any action on the website except for searching and retrieving information. Make sure the tasks include the phrase "on bbc.com" so that the agent searches for the bbc.com website.

## F    PROMPT USED FOR EVALUATING CAPTCHA ANALYSIS AGENT TRACES

**Model Used:** o4-mini

**System Prompt:** You are an LLM judge evaluating the output of an agent attempting to traverse different websites to solve a task. The task involves navigating Expedia, a travel website which often uses captchas to block robots from traversing the website. Your goal is to identify the captcha avoidance strategies used by the agent. The list of captcha avoidance strategies are as follows:

1. `cache` - attempts to use Google's cache to retrieve an unblocked version of the website.
2. `mobile` - attempts to use the mobile version of the website to retrieve an unblocked version of the website.
3. `direct_link` - attempts to open the website by navigating directly to a link that may have the correct website instead of searching for it on Google.
4. `google_search - attempts to conduct a Google search to identify alternative links to the same website (without using any cache terms - if the Google search has cache terms, then the `cache` strategy was used).
5. `randomized_interaction` - attempts to wait random amounts of time before completing an interaction to circumvent bot detection algorithms.
6. `reloads` - reloads the website in an attempt to remove the captcha.
7. `new_tab` - attempts to open the website in a new tab to avoid any session cookies being associated with its search.
8. `switch_websites` - switches to a non-Expedia website to solve the task instead of trying to navigate to Expedia.
9. 'internal_navigation' - attempts to go to the home webpage of Expedia, and navigate to the correct webpage from the home webpage.
10. `country_domain` - attempts to use a different country domain of Expedia to retrieve an unblocked version of the website.
11. `text-only rendering` - attempts to perform a text-only render or retrieve the plaintext version of the website by using a proxy such as Textise.
12. `public proxy` - attempts to use a public proxy such as AllOrigins to avoid bot protection mechanisms.
13. `internet_archive` - attempts to use the Internet Archive's Wayback Machine to retrieve an unblocked version of the website.
14. `google_travel_integration` - attempts to use the Google Travel integration to solve the task. This strategy is only applicable if the term "Google Flight Integration" or "Google Travel Integration" is mentioned in the agent's output.

You should output a JSON string in the format `strategy_name`: `bool` which has all the keys as strategy_names and the values as "true" or "false" depending on whether the strategy was detected in the agent output. For example, if only the `reloads` and `new_tab` strategies were detected in the output, your output should be:
```

{
"cache": false,
```

756 "mobile": false,
757 "direct_link": false,
758 "google_search": false,
759 "randomized_interaction": false,
760 "reloads": true,
761 "new_tab": true,
762 "switch_websites": false,
763 "internal_navigation" false,
764 "country_domain": false,
765 "text-only rendering": false,
766 "public_proxy": false,
767 "internet_archive": false,
768 "google_travel_integration": false
769 }
770 ```

771 Make sure that all the keys are present in the output, even if their value is false. Only output the
772 JSON string, do not include any other text, and make sure that the only keys in the JSON string are
773 the ones listed above. Do not include any additional keys or values in the output.

774 **User Prompt:** {Agent Trace}

## G PROMPT USED FOR EVALUATING POP-UP BANNER AGENT TRACES

**Model Used:** o4-mini

**System Prompt:** You are an LLM judge evaluating the output of an agent attempting to traverse
different websites to solve a task. The task involves navigating to a website with a cookie banner,
privacy pop-up, or pop-up ad and completing the assigned task. You are provided with the task and
the agent's output, and your goal is to identify if the agent was successful in closing the cookie banner
or pop-up ad and progressing with the task.

You should output a JSON string in the format {"banner_detected": `bool`, "banner_closed": `bool`,
"task_successfully_completed": `bool`} where the "banner_detected" key indicates whether the
agent found a cookie banner, privacy pop-up, or pop-up ad at any point while trying to complete the
task, and the "banner_closed" key indicates whether the agent successfully closed it. If no cookie
banner or pop-up ad was detected, both values should be false. The "task_successfully_completed"
key should be set to True if the agent states it successfully completed the task at the end of the trace.

Make sure that all the keys are present in the output, even if their value is false. Only output the
JSON string, do not include any other text, and make sure that the only keys in the JSON string are
the ones listed above. Do not include any additional keys or values in the output.

**User Prompt:** {Agent Trace}

## H PROLIFIC USER STUDY COMPENSATION

We collect tasks from users in 3 batches: the 6 participants in the first batch of the pilot study were
paid $1.50 per response based on a projected median response time of 11:00 minutes for an hourly
cost of $8.17/hr. The second and third batches were paid at an hourly rate of $8.01/hr, with the 28
responses in the second batch paid at a rate of $5.40 per response based on the calculated median
response time of 40:28 minutes, while the 179 responses in the third batch were paid at a rate of
$4.69 per response based on the calculated median response time of 35:09 minutes.

## I  FAILURE MODE DISCOVERY DETAILS

### I.1  DATASET FEATURIZATION

We apply *Dataset Featurization* (Bravansky et al., 2025) to surface common failure modes from our step-level agent task labels, following the unsupervised, two-stage pipeline of (i) feature proposal via contrastive data–reconstruction prompts and (ii) forward selection under a reconstruction–perplexity objective. Concretely, for each target goal string $x$, we draw $C=5$ contrastive strings $\{r_c\}_{c=1}^5$ from the corpus and prompt GPT-4o to propose $K=4$ short ($\leq 20$ words) binary predicates that are true of $x$ while (ideally) not holding for the $\{r_c\}$. This contrastive step forces candidates to be discriminative rather than generic. Pooling across $N=218$ goal–feedback examples yields $872$ initial feature hypotheses. We embed each candidate (and associated step text) with `text-embedding-3-small`, standardize embeddings, and perform K-means with target granularities chosen to achieve interpretable coverage yielding $15, 10, 5$ clusters across sweeps. From each cluster we retain one representative phrasing. We then assign binary truth values by asking GPT-4o (temperature $= 0$) to evaluate every (goal string, clustered feature) pair, producing a $N \times K'$ binary matrix (labels "Y/N").

The final failure modes are selected from these clusters by testing how well they allow a language model model (Llama-3-8B) to reconstruct the step-by-step labels. Namely, we treat active features for a text as a newline-delimited context and compute mean per-text perplexity

$$\mathrm{PPL}(D \mid \phi) \;=\; \frac{1}{N} \sum_{n=1}^{N} \mathrm{PPL}\big(x^{(n)} \,\big|\, \mathtt{ctx}(\phi(x^{(n)}))\big),$$

then greedily append the feature $F$ that most reduces perplexity, i.e.,

$$F \;=\; \arg\min_{F'} \; \mathrm{PPL}\big(D \,\big|\, \phi \cup \{F'\}\big),$$

stopping when no candidate yields a further drop (or a feature budget is reached). Following DF, we use a static reconstruction prompt and cache log-probabilities for texts where a feature evaluates to FALSE to avoid redundant computation. The resulting cluster summaries instantiate the final failure modes.

### I.2  DOCENT

We also use an API-only method, Docent Meng et al. (2025), to help confirm the consistency of our clusters and summaries across featurization methods. We pass the human-step level labels of each agent goal, along with the prompt: `Based on the step-by-step feedback metadata on each agent step, find the failure modes where the agent fails to complete research tasks.  Be granular, e.g.  not just "failure" but "failure due to the agent not properly handling x in case y.".` The failure modes are displayed in Figure 4(a); we find significant overlap between our dataset featurization failure modes and the Docent failure modes. To featurize the dataset, Docent uses Claude Sonnet 4 to produce natural language summaries of our human step-level labels, two of these (for the cookie and captcha failure modes) are presented in Figure 4(b) and (c).

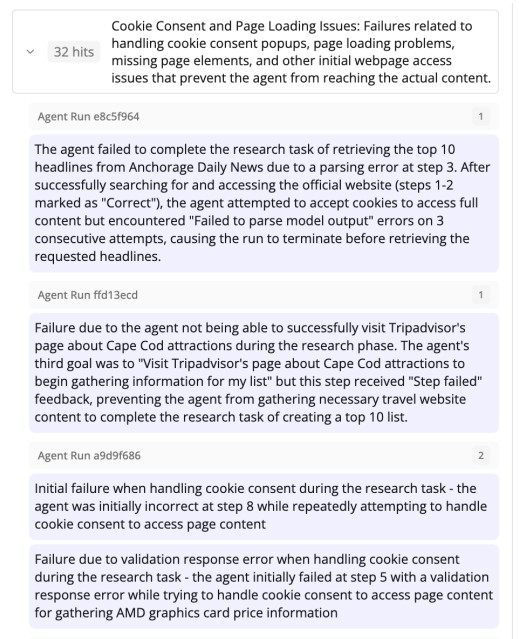

> 85 hits — Model Output Parsing Failures: Technical failures where the agent cannot parse or process its own generated output, preventing execution of research steps due to JSON parsing errors, malformed responses, or inability to interpret structured data.

> 29 hits — Website Access Blocking: Failures caused by anti-bot security measures including CAPTCHAs, Cloudflare verification, human verification checks, throttling errors, and other website protection mechanisms that prevent the agent from accessing research sources.

> 85 hits — Navigation and UI Interaction Errors: Failures in basic web navigation including inability to click elements, scroll properly, handle dropdowns, navigate between pages, or interact with specific UI components needed to complete research tasks.

> 32 hits — Cookie Consent and Page Loading Issues: Failures related to handling cookie consent popups, page loading problems, missing page elements, and other initial webpage access issues that prevent the agent from reaching the actual content.

> 90 hits — Search Execution and Query Formulation Problems: Failures in executing search queries, formulating proper search terms, triggering search results, or handling search interface interactions across various platforms.

> 102 hits — Data Extraction and Content Processing Failures: Failures in extracting specific information from successfully accessed webpages, including inability to parse tables, extract headlines, process lists, or gather required data points from loaded content.

> 81 hits — Task Scope and Planning Errors: Failures due to inadequate task planning including checking only one website when multiple were required, incomplete goal specification, incorrect task scoping, or failure to execute planned research steps entirely.

> 110 hits — Information Filtering and Verification Problems: Failures in applying proper filters (date ranges, categories, criteria), verifying data accuracy, distinguishing relevant from irrelevant results, or extracting information that meets specific requirements.

> 12 hits — Date Selection and Form Input Errors: Specific failures related to inputting dates, selecting time ranges, handling calendar interfaces, filling out travel or booking forms, and managing temporal data requirements.

> 72 hits — Final Synthesis and Completion Issues: Failures in the final stages of research tasks including inability to synthesize collected data, provide summaries, format final outputs, or complete the overall research objective despite successful intermediate steps.

> 7 hits — Residuals

(a) Docent summarization of our human step-level labels

---

⌄ 32 hits — Cookie Consent and Page Loading Issues: Failures related to handling cookie consent popups, page loading problems, missing page elements, and other initial webpage access issues that prevent the agent from reaching the actual content.

**Agent Run e8c5f964** — 1

The agent failed to complete the research task of retrieving the top 10 headlines from Anchorage Daily News due to a parsing error at step 3. After successfully searching for and accessing the official website (steps 1-2 marked as "Correct"), the agent attempted to accept cookies to access full content but encountered "Failed to parse model output" errors on 3 consecutive attempts, causing the run to terminate before retrieving the requested headlines.

**Agent Run ffd13ecd** — 1

Failure due to the agent not being able to successfully visit Tripadvisor's page about Cape Cod attractions during the research phase. The agent's third goal was to "Visit Tripadvisor's page about Cape Cod attractions to begin gathering information for my list" but this step received "Step failed" feedback, preventing the agent from gathering necessary travel website content to complete the research task of creating a top 10 list.

**Agent Run a9d9f686** — 2

Initial failure when handling cookie consent during the research task - the agent was initially incorrect at step 8 while repeatedly attempting to handle cookie consent to access page content

Failure due to validation response error when handling cookie consent during the research task - the agent initially failed at step 5 with a validation response error while trying to handle cookie consent to access page content for gathering AMD graphics card price information

---

⌄ 29 hits — Website Access Blocking: Failures caused by anti-bot security measures including CAPTCHAs, Cloudflare verification, human verification checks, throttling errors, and other website protection mechanisms that prevent the agent from accessing research sources.

**Agent Run 320c76f4** — 1

Failure due to the agent repeatedly failing to properly handle Google CAPTCHA verification challenges when attempting to search for TypeRacer website, with 5 consecutive incorrect attempts at CAPTCHA solving preventing successful navigation to the research target.

**Agent Run f9801f92** — 1

Failure due to the agent not properly handling API throttling errors when attempting to access Amazon's official Best Sellers list. The agent encountered a "Throttling error" at step 6 when trying to "Resolve throttling error and retrieve official Best Sellers in Fiction", and despite attempting remediation through page refresh and content re-extraction in steps 7-8, the throttling error persisted, preventing successful completion of the research task to find current best-selling fiction books with their prices and ratings.

**Agent Run 273a8143** — 1

Failure due to the agent not effectively handling anti-bot measures during final content extraction attempts. The emergency scroll and extract strategy resulted in incomplete and scrambled product information, with extraction ultimately failing due to CAPTCHA challenges and content blocks preventing access to the trending home décor listings.

(b) Example individual datapoint captioning (in blue), from the Claude Sonnet 4, for the failure mode dealing with cookies.

(c) Example individual datapoint captioning (in blue), from the Claude Sonnet 4, for the failure mode dealing with cookies.

Figure 4: Failure mode identification with Docent (Meng et al., 2025).

Table 2: Agent failure modes found form the human step-level labels via dataset featurization Bravansky et al. (2025), under different granularity $k$. Bolded rows correspond to the failure modes we explore in detail via generated tasks in Section 5.

| $k$  Failure mode | Count | Share (% of total 220) |
|---|---|---|
| **$k = 5$** | | |
| Complex tasks with multiple steps | 185 | 84.9 |
| **Navigation to specific website sections** | 116 | 53.2 |
| Straightforward task sequences | 65 | 29.8 |
| Repeated parsing errors | 54 | 24.8 |
| Task completion execution errors | 22 | 10.1 |
| **Cookie consent handling failures** | 12 | 5.5 |
| **$k = 10$** | | |
| Specific list extraction tasks | 148 | 67.9 |
| **Direct URL navigation attempts** | 77 | 35.3 |
| Goal completion without failures | 68 | 31.2 |
| Repeated parsing errors | 63 | 28.9 |
| Concise task structure | 58 | 26.6 |
| High frequency unsuccessful attempts | 57 | 26.1 |
| Technical errors (non-navigation) | 44 | 20.2 |
| Product category focus | 37 | 17.0 |
| **Cookie consent success** | 20 | 9.2 |
| Inadequate human feedback | 8 | 3.7 |
| **$k = 15$** | | |
| **Navigation to specific sections** | 157 | 72.0 |
| Repeated task completion attempts | 109 | 50.0 |
| Parsing failure feedback | 99 | 45.4 |
| Concise task structure | 79 | 36.2 |
| Detailed extraction from tables | 74 | 33.9 |
| Goal completion without errors | 67 | 30.7 |
| Multiple information location attempts | 65 | 29.8 |
| Repeated parsing errors | 51 | 23.4 |
| Technical errors (non-navigation) | 49 | 22.5 |
| URL error references | 44 | 20.2 |
| Task completion execution errors | 37 | 17.0 |
| Travel-related task focus | 36 | 16.5 |
| **Cookie consent success** | 21 | 9.6 |
| **CAPTCHA/verification failures** | 15 | 6.9 |
| Inadequate human feedback | 8 | 3.7 |

## J CAPTCHA SOLVING STRATEGY PREFERENCES

| Captcha-Solving Strategy | Gemini 2.5 | o4-mini | R1 | Llama-4 | Claude-3.7 |
|---|---|---|---|---|---|
| Cache | 0.00 | 45.45 | 1.82 | 0.00 | 0.00 |
| Mobile | 0.00 | 58.64 | 5.91 | 0.00 | 0.00 |
| Direct Link | 25.45 | 97.73 | 81.82 | 69.55 | 60.91 |
| Google Search | 42.73 | 100.00 | 94.55 | 61.36 | 77.27 |
| Randomized Interaction | 0.00 | 0.45 | 25.91 | 3.18 | 0.00 |
| Reloads | 9.09 | 3.64 | 27.73 | 12.27 | 1.82 |
| New Tab | 4.55 | 60.45 | 52.27 | 69.55 | 12.27 |
| Switch Websites | 15.45 | 5.00 | 31.82 | 22.73 | 60.00 |
| Internal Navigation | 0.91 | 40.00 | 22.73 | 0.00 | 0.45 |
| Country Domain | 0.45 | 29.09 | 0.91 | 0.00 | 0.00 |
| Text-only Rendering | 0.00 | 7.27 | 0.00 | 0.00 | 0.00 |
| Public Proxy | 0.00 | 1.36 | 0.00 | 0.00 | 0.00 |
| Internet Archive | 0.00 | 3.64 | 0.00 | 0.00 | 0.00 |
| Google Travel Integration | 0.00 | 0.45 | 1.36 | 0.00 | 0.91 |

Table 3: Percentage of times a particular captcha avoidance strategy was deployed by a model while solving tasks in the Expedia task dataset

## K  POP-UP BANNER CLOSURE SCENARIOS

| Pop-Up Banner Scenarios | Gemini 2.5 | o4-mini | R1 | Llama-4 | Claude-3.7 |
|---|---|---|---|---|---|
| Banner Detected | 53.75 | 91.25 | 0.00 | 98.75 | **100.00** |
| Banner Closed | 4.65 | **17.81** | 0.00 | 17.72 | 7.5 |
| Marked as Completed | 7.5 | 23.75 | **53.75** | 3.75 | 2.5 |

Table 4: Percentage of times a particular pop-up banner scenario was observed in an agent's trace while executing tasks from the BBC task dataset. We note that the percentage in the Banner Closed row is determined with respect to the number of tasks where the agent determines that there is a banner as per the LLM judge. The other two rows (Banner Detected and Marked as Completed) are computed with respect to the total number of tasks in the BBC dataset.

## L  DIRECT NAVIGATION ACTIONS TAKEN

| | Google API | Google Site | Wiki | Direct Answer | Failed |
|---|---|---|---|---|---|
| Claude-3.7 | 97 | 3 | 0 | 0 | 0 |
| R1 | 98 | 2 | 0 | 0 | 0 |
| Gemini 2.5 | 50 | 0 | 0 | 0 | 50 |
| Llama-4 | 74 | 26 | 0 | 0 | 0 |
| o4-mini | 76 | 2 | 1 | 9 | 12 |

Table 5: Count of times agents taking different actions when asked questions from TriviaQA dataset.

## M  EXAMPLES OF USER-SUBMITTED TASKS

**Task Prompt 1:** Find me the last available train from Cardiff Central to Barry Docks station today on trainline.

**Task Prompt 2:** Compare flight prices from washington DC to Paris france to flights from washington DC to Kyoto Japan

## N    AGENT ACTION CLUSTERING

| Browser Action Type | Percentage of total actions |
|---|---|
| Search-Based Information Gathering (search_google, click_element_by_index, extract_page_content) | 24.3% |
| Direct Website Navigation (go_to_url, extract_page_content) | 12.3% |
| Form Interaction and Input (input_text, click_element_by_index, click_element_by_xpath, click_element_by_selector) | 20.7% |
| Multi-Tab Browsing (switch_tab, close_tab, open_url_in_new_tab) | 3.6% |
| Page Navigation and Scrolling (scroll_down, scroll_up, go_back, scroll_to_text) | 7.2% |
| Advanced Browser Controls (wait, wait_for_element_to_be_visible, send_special_keys, get_dropdown_options, select_dropdown_option_by_text, drag_and_drop) | 1.1% |
| Misc (e.g. Content Preservation, Task Completion) | 30.7% |

Table 6: Browser actions across all agent traces. Traces are labeled and clustered with the Docent API (Meng et al., 2025) using GPT-5.

## O  TASK CLUSTERING

| Cluster Label | Percentage |
|---|---|
| **News headlines and current events** 
 Tasks requesting today's top headlines, current news stories, or trending articles from specific news websites or media outlets. | 10.5% |
| **Product price comparison across retailers** 
 Tasks that involve comparing prices of specific products (laptops, phones, TVs, etc.) across multiple e-commerce sites or retailers. | 2.9% |
| **Flight and travel booking comparison** 
 Tasks focused on finding and comparing flight prices, hotel rates, train schedules, or other travel services across booking platforms. | 6.6% |
| **Top-ranked lists and rankings** 
 Tasks requesting ordered lists like "top 10" or "best of" items such as movies, restaurants, players, or products with specific criteria. | 32.5% |
| **Sports statistics and player information** 
 Tasks seeking sports-related data including player stats, team performance, game results, or sports rankings and analysis. | 5.9% |
| **Entertainment content discovery** 
 Tasks about finding movies, TV shows, music, books, or games with specific criteria like ratings, genres, or popularity. | 8.8% |
| **Product specifications and reviews** 
 Tasks requesting detailed technical specs, features, or expert reviews for specific products or devices. | 4.9% |
| **Recipe and cooking information** 
 Tasks seeking cooking recipes, ingredients, or food-related content from cooking websites or food platforms. | 1.5% |
| **Real estate and accommodation listings** 
 Tasks involving finding properties, hotels, lodges, or rental accommodations with specific criteria and pricing. | 2.7% |
| **Weather and location-specific information** 
 Tasks requesting current weather conditions, forecasts, or location-specific data and information. | 4.6% |
| **Financial and market data** 
 Tasks seeking information about stock prices, cryptocurrency trends, economic indicators, or financial market analysis. | 1.5% |
| **Educational content and summaries** 
 Tasks requesting explanations, summaries, or educational content about specific topics, concepts, or historical information. | 11.0% |
| **Event listings and schedules** 
 Tasks seeking information about upcoming events, concerts, shows, or scheduled activities with dates and pricing. | 2.2% |
| **Transportation schedules and routes** 
 Tasks about train times, public transit schedules, or specific transportation route information and pricing. | 4.4% |

Table 7: Task clusters, from the Docent API (Meng et al., 2025) using GPT-5 (medium reasoning effort).

## P  QUALTRICS SURVEY QUESTIONS

In this section we provide the questions used to collect tasks and step-level feedback from platform users.

### P.1  TASK DESIGN QUESTION

The goal of this survey is to understand how well AI can perform everyday web browsing tasks, so you will be asked to enter a task involving searching and navigating through websites. AI agents

will then try to execute the task and return the results to you. We ask you to enter tasks that involve clicking and interacting with different parts of various websites (which we call "**interactive tasks**") and NOT just a simple Google search query (which we call "**search tasks**") where the answer is easily provided by Google, is in the texts and links alongside search results, or is an open-ended question that a chatbot can answer without clicking on any website.

An example of an interactive task is: "Give me a summary of the featured article on Wikipedia's homepage". This is because a human trying to do the task would need to go to Wikipedia's website, identify the featured article, click on it to read the complete article, and then summarize the information. An example of a search task is: "What is the capital of France?". A human trying to do this task would Google the answer, and Google would present the exact answer in a box, as well as in the links and short summary text under each search result.

Here are some examples of **valid interactive tasks**, which involve interacting with specific websites and clicking on them: 1. What are today's top 20 headlines from CNN? 2. Compare the bus prices for one-way tickets from Boston to New York next Saturday on different ticket purchasing websites. 3. Create a list of the top-ranked chess players on chess.com from Belgium.

Here are some examples of **invalid search tasks (and why they are invalid)**:

1. How do I increase my concentration while working? *This is invalid because it can be answered using a chatbot and does not require clicking on a specific website.*

2. What is the weather today? *Google will output this answer in a box displayed at the top of search results, again does not require clicking on a specific website*

3. Who are the members of the Beatles? *Google provides a lot of links with the text containing the answer to this question under those links, so you do not need to click on a website to answer this question.*

Please only submit interactive tasks (**search tasks will be counted as INVALID submissions and rejected on Prolific**), and **do not resubmit the example tasks** we described above (we already have them). To make sure your task is counted, check that Googling your search query requires at least one click on a website link to successfully complete the task.

Currently, our system cannot interact with websites that require login IDs and passwords, so **do not submit any tasks requiring logging into a website**. Do not enter any sensitive information (such as usernames, passwords, or personal or financial information of any kind) at any point, either on this form or in the web app interface with the agent. We will only ask you for your Prolific IDs on this form and the web app for payment and cross-verification of answers. Also please avoid submitting tasks involving controversial or political topics, or questions about specific individuals - it is very likely that the system will reject tasks on those topics and you will be unable to complete the survey.

In the text box underneath, enter the description of the interactive task that is to be submitted to the AI agent. Your task must be in English. This is for our own records - you will submit the task to the actual AI agent on another website. Before submitting the answer to this question, copy the text you have entered here. We will check to make sure that the text entered here is identical to the text submitted to the agent.

P.2   QUESTION ID

Now, go to `<website link>`. Make sure that you are on the Arena (battle) tab - you can change tabs by clicking through them on the top. Only responses submitted in the Arena (battle) tab will be counted. Do not close this website until you have completed the rest of this Qualtrics form - you will need to refer to it to answer the remaining questions.

Scroll down to where it says "Enter your prompt". Paste the task description you submitted in Question 1 in the box. Next to that box there is a User ID box - enter your Prolific ID there. Next to that, there is a Task ID box, for which you should create a unique description (like "wikipedia_1") to differentiate it from any other tasks you submit. You can rephrase the language of a previously submitted task to make the task description more specific or more general. We ask that if you are rephrasing a previous task, please reuse the task ID so that we know that you are rephrasing a previous task. You are permitted to submit at most 3 versions of the same task with the same task ID.

Before pressing the Send button, enter the task ID you are submitting as the answer to this question.

## P.3 ERROR CHECK

Now press the Send button. **Make sure that you submit the task on `<website link>` to ensure that your response is recorded - if we do not see a submission on that website, this Qualtrics form submission will be considered invalid and will be rejected on Prolific.**

The task has been sent to two AI agents - you will be asked to rank their outputs. It might take a while for the agents to generate their responses. If after 20 minutes, no response has been generated, or you see only the word "Error" on **both** panes, please end the survey here and let us know. **Note: If you submit the form before 20 minutes and tell us that no text was generated, we will count the response as invalid because it might take a while for the response to be generated when there are many users using the app.**

Please note that you should continue the survey if any text other than the word "Error" was generated - if the agent fails to do the task or says that it failed to do a part of the task, you will be able to tell us that in a later section. If only one pane shows an error, please continue with the form and select Yes - you can tell us about the error at a later point in the survey. **If the word "Step" or "Failure" appears in the output, please continue the survey - it is important for us to know which specific step number failed.** If we determine that the model produced a step or a specific failure in the output and No output was selected as an answer to this question, we will be unable to accept this response.

**If you select No output here, it will end the survey, and any subsequent attempts to retake the survey will be considered invalid. We will check with your prompt to make sure that we can reproduce any errors to verify this survey submission.**

Were any outputs produced by the AI agents in the grey textbox?

## P.4 LEFT AGENT CORRECT STEP GENERATION CHECK

We check that the agent generated steps in the correct format, as shown in Figure 5.

Both panes for both AI agents should have generated some text demonstrating the steps and actions taken by the AI agent.

An example of the text that might be generated is as follows:

🧠 Starting an agent with main_model=anonymized, planner_model=anonymized, extraction_model=anonymized

🚀 Starting task: What is the capital of France?
📍 Step 1
👍 Eval: Unknown - Starting fresh session with blank page
🧠 Memory: Initiating Google search for the answer. Step 1/15 completed.
🎯 Next goal: Retrieve capital of France from search results 🛠 Action 1/1: {"search_google":{"query":"capital of France"}} 🔍 Searched for "capital of France" in Google Can you clearly identify the steps in at least one of the agent responses, or is there some other text produced?

For the agent on the **left side**, does the generated text have clear steps like this, or was the output some very differently formatted text? By clear steps, we mean that you should be able to clearly identify where one step ends and the next step begins because each step will have a line starting with "📍 Step" at its start.

Figure 5: In this step (and all following steps), for the questions for the right agent, simply replace all occurrences of the word "left" with "right".

## P.5 LOG ID COLLECTION

Scroll to the very end of the **left-side** agent's output. It will have produced a log ID, which the text after the phrase: "Log ID: ". Copy the log ID and paste it here. If no log ID was generated, enter "N/A". The log ID will be used to verify that this query was submitted on the BrowserArena website,

so please fill this field accurately to avoid your response being marked as invalid and rejected on Prolific.

## P.6 LEFT AGENT STEP-LEVEL FEEDBACK

We present the complete step-level feedback question in Figure 6.

Now, examine the output of the **left-side** AI agent. There are several steps, each of which should describe a "Next Goal" and an "Eval". The Eval sentence is the agent's attempt to evaluate whether it achieved the goal it had set for itself in the previous step. Your next task is to provide a human evaluation of whether the agent correctly performed each step. If the agent said that it will visit a website in a previous step but is unable to click on the link to get there in this step, mark this step wrong. If the agent said that it will collect some information in a previous step, but collects incorrect information in this step, mark this step wrong. There is a GIF generated at the bottom of the agent's output that should show the agent visiting different websites and clicking on different links - use it to see exactly what the agent was doing.

The _total number of steps_ for the **left-side** AI agent is the largest number X which appears in the output of the model as " 📍 Step X". For example, the output may contain 5 lines (lines in between the steps have been replaced by ellipses):

📍 Step 1
...
📍 Step 2
...
📍 Step 3
..
📍 Step 3
...
📍 Step 4
...
Result:

In this example, note that the text " 📍 Step 3" is repeated. Despite that, the _total number of steps_ in this example is 4, since it is the largest number appearing next to " 📍 Step ". The maximum number of possible steps is 15.

You will now be asked to provide feedback about each step. **Please make sure to provide feedback for each individual step and not for all the steps in one box - if feedback is missing for individual steps the response will be rejected.** If the step was correctly performed according to you, simply enter the word "Correct". If it was performed incorrectly, explain why you think the step was incorrect. Sometimes, the agent will produce multiple lines that have the same step number i.e. multiple lines "Step 3" in the previous example. This is because the agent tried to execute Step 3, failed, and then retried. In that case, combine your feedback for all lines with the same step number, i.e. put all your feedback for those steps in the box for Step 3. If this form is asking you about feedback for a step larger than the _total number of steps_, write "N/A". In the previous example, since the total number of steps is 4, all steps from 5-15 should have "N/A" in their boxes. In this example, steps 1-4 should not have "N/A" - they should either have "Correct" or your own feedback on the step.

Here are some examples of questions whose answers may help you provide feedback for individual steps: did the agent search for the wrong term, or visit the wrong website? Was it unable to click on a particular part of the website? Did the agent generate text that the system could not translate into an action to take on the website?

Please do not write anything except "N/A" for any step numbers beyond the _total number of steps_ in the agent's output. We will cross reference these answers with the agent outputs and it is important that feedback is provided only for the steps that the agent generates.

In the below text box, enter your feedback for Step 1. Enter "Correct" if the step was correctly executed, provide your feedback if the step was incorrectly executed, and enter "N/A" if it is larger than the total number of steps.

Figure 6: The last two lines of this question ("In the below...number of steps") are repeated for a total of 15 steps each for the left and right agent outputs.

## P.7 VOTING

Now, on the website, vote for the best response between the two models by clicking the **Left**, **Right**, or **Tie** buttons there. Then, enter your vote here as "Left", "Right", or "Tie". Use your best judgement to analyze which model did better on your task - if both models did not succeed in completing the task, vote for the model that got the closest to completing the task. **Do not click on the button marking both responses as bad - the goal is to evaluate which model gets closest to completion.**

