# OpenReview forum: "BrowserArena: Evaluating LLM Agents on Real-World Web Navigation Tasks"
_ICLR.cc/2026/Conference — Submitted to ICLR 2026_

### Official Review · Reviewer_G5Uu · 2025-10-30

**Soundness:** 1
**Presentation:** 1
**Contribution:** 1
**Rating:** 2
**Confidence:** 4

**Summary:**

This work builds a toolkit for collecting web-based tasks and annotating agent playing trajectories in an open web environment. The authors also conduct several experiments to analyze the performance of current large models on the open web, and identify their main failure patterns.

**Strengths:**

This is a toolkit that allows web agents to execute tasks in the open web environment, making it easier to crowdsource more tasks and annotations from users. Compared with contributions focusing on datasets or benchmarks, this work is more suitable to be evaluated under a demo track.

**Weaknesses:**

First, this data collection toolkit should ideally address at least some of the failure patterns it identifies, such as handling CAPTCHA and closing pop-ups. These are not really the intended goals of web-agent research — these are trivial, procedural problems that can be solved with simple dedicated pipelines. For example, we can’t really say that solving CAPTCHA is a core capability which web agents are developed for. If that’s the case, why doesn’t the toolkit itself handle these issues to avoid their interference with the main conclusions?
(It feels somewhat ironic if the biggest challenge revealed by this platform for current web agents turns out to be “solving CAPTCHAs and closing banners.”)

Second, before submission, the authors could benchmark their platform against existing “arena” frameworks such as Chatbot Arena, and evaluate which features an arena should have—for example, a more fine-grained ranking system and anomalous user detection. These design elements are crucial for ensuring the accuracy and robustness of evaluations in an open arena setting.

Third, has this arena attracted a large user base? If not, what is the plan to engage more users? Designing and releasing an open-source toolkit is primarily an engineering task, but to demonstrate that the toolkit truly works, it needs to be instantiated with a sufficient number of real data points, showcasing its usage and analysis at scale.

**Questions:**

* Why doesn’t this toolkit help mitigate issues like CAPTCHA and pop-ups to prevent them from distorting the main evaluation conclusions?
* How does the arena ensure evaluation accuracy and robustness in an open setting?
* Has it attracted a large number of users, and if not, what strategies are in place to do so?

---

> ### Author Response · Authors · 2025-11-21
> **Response (1/1)**
>
> Thank you for your feedback, and for recognizing our contributions to scaling data collection in an open-web environment.
>
> **Regarding Custom Tools for CAPTCHA Removal and Pop-Up Banner Closure**
>
> While we agree that it is possible to implement custom tools to handle specific failure modes, the reason we do not implement failure-mode-specific fixes is that the contribution of our paper is not to highlight engineering limitations of data collection platforms (e.g., that web agents fail at captcha resolution). Our contribution involves the evaluation methodology: we argue that collecting user preference data and user-provided step-level feedback based on LLM-generated step goals helps surface recurring failure scenarios that can be more comprehensively studied. We select captcha removal, pop-up banner closure, and direct navigation as failure modes that we can study in greater detail because they can be repeatedly reproduced on our platform. As a result, we can synthetically generate tasks to analyze statistical differences in how different LLMs handle these failure modes. The core difference between our evaluation methodology and other ground-truth-based web agent benchmarks is that these failure modes would be filtered out in those benchmarks due to missing ground truth. Additionally, since task success is tied to specific ground truths in those benchmarks, it is extremely difficult to measure quantities like “diversity of strategies employed in a failure scenario”, which is easily facilitated by our approach.
>
> Finally, in addition to captcha resolution and pop-up banner closure, our approach also surfaces scenarios like “direct navigation to specific URLs”, “date selection issues”, “issues handling cookie consent”, etc. While it may be possible to engineer the system to mitigate or reduce specific pitfalls, we highlight these issues as evidence of failure modes that would not be easily detectable in a ground-truth-based benchmark, which are not simply artifacts of the data collection platform used.
>
> **Regarding Comparison to ChatbotArena**
>
> Our codebase for collecting data is built on top of the Chatbot Arena codebase (Line 179). As a result, we have the same fine-grained ranking system and anomalous user detection system as ChatbotArena. Could you please elaborate on which exact quantities you would like us to benchmark?
>
> **Regarding Attracting a Large User Base**
>
> While we believe scaling our benchmark to a true arena is an important direction for our work to have a practical impact, our main contribution is the design of this arena, and scaling it is beyond the scope of an academic contribution. We are exploring the possibility of collaborating with companies to help with the engineering effort required for scaling.
>
> We have relied on paying users for the data collected for this study on Prolific and used the initial data to construct our leaderboard. Additionally, we plan to present our work at workshops at machine learning conferences to raise awareness about the project. Finally, we are also having conversations with evaluation and web agent companies to scale the benchmark with their support.

---

### Official Review · Reviewer_jXW9 · 2025-11-01

**Soundness:** 2
**Presentation:** 3
**Contribution:** 2
**Rating:** 4
**Confidence:** 3

**Summary:**

The paper introduces BrowserArena, a live, open-web platform for evaluating LLM web agents using user-submitted tasks and Chatbot Arena-style pairwise comparisons. To address limitations of final-output metrics, the core contribution is a methodology utilizing granular step-level human feedback collected on agent traces. Analyzing 109 user-submitted tasks, the authors identify three consistent agent failure modes: captcha resolution, pop-up banner removal, and unintended direct navigation. Subsequent targeted experiments demonstrate notable behavioral differences and brittleness across contemporary LLM agents in handling these real-world obstacles.

**Strengths:**

The most significant contribution is the diagnostic evaluation methodology rooted in collecting step-level human annotations, which effectively surfaces intermediate performance issues that traditional final-output benchmarks overlook. The use of live, user-submitted tasks enhances the ecological validity, avoiding the highly specific or artificial constraints of self-hosted environments. Furthermore, the empirical rigor shown by using the collected feedback to identify and construct specialized datasets focused on persistent failure modes (e.g., Captcha, Pop-up) provides quantifiable insights into agent deficiencies.

**Weaknesses:**

1. The participants' motivation must be considered when building this arena. The motivation of participants in a chatbot arena is clearly stronger, as the inherent instability of LLMs makes people need to see the outputs of different LLMs, and the overall process is fast and efficient. However, for GUI Agents, participants seem to lack sufficient motivation to interact by watching two GUI Agents output different reasoning. Without enough motivation, this system will not be scalable, and the value will not be significantly different from offline labeled evaluation data. I believe this is the core problem of this work. To justify this motivation, it should be demonstrated either through the actual number of user-generated interactions or by conducting interaction experiments in the field of HCI.
2. Some insights arise from the limitations of the evaluation setup: for instance, the lack of methods and evaluation for multimodal observation and grounding.
3. Relying on the browseruse framework is also unreasonable because there are multiple implementation paradigms for web agents, which significantly limits the value generated by the evaluation.

**Questions:**

1. How to evaluate the validity of step-level feedback? This is because many tasks for GUI Agents cannot be judged as correct or incorrect based solely on the current step (as a single semantic unit/subtask often requires multiple steps), and some GUI Agents may be skilled at recovering from errors.
2. What specific distributional insights are there regarding the user-submitted queries, and what potential biases exist?

---

> ### Author Response · Authors · 2025-11-21
> **Response (1/3)**
>
> Thank you for your feedback, and for recognizing that we surface issues overlooked by other benchmarks.
>
> **Regarding User Interest**
>
> First, while we believe scaling our benchmark to a true arena is an important direction for our work to have practical impact, our main contribution is the design of this arena, and scaling it is beyond the scope of an academic contribution. We are exploring the possibility of collaborating with companies to help with the engineering effort required for scaling.
>
> Second, based on our median survey completion times, we observe high levels of engagement, with an average median survey completion time of 35:10 minutes across three batches of data collection (reported in Section 4.1 and Appendix H). Additionally, given that we ask users to mark steps as either “correct” or provide a description of why they feel the step was executed incorrectly, we can also examine the mean length of the reasoning provided for incorrect steps to assess the thoroughness of the feedback. The mean length of justification for incorrect steps provided was 14.96 words, with a median of 12 words, suggesting that users provide sufficiently detailed feedback per step.
>
> We envision two potential paths to scaling our approach to a larger pool of participants, with varying tradeoffs. The first, which we refer to as the “one-step approach”, is the format used for collecting data in the paper, where users submit tasks to two randomly selected LLMs and vote for their preferred output. This approach is consistent with existing arena-style benchmarks where users evaluate LLM performance on their own tasks. While it may take longer to generate the trajectory due to multiple calls to the LLM, we believe that actual user interaction time will not significantly differ from other arenas involving long test-time computation (such as WebDev Arena, which requires models to design a website that is then rendered).
>
> The second approach, which we refer to as the “two-step” approach, involves collecting tasks from users separately from the votes. Users submit tasks, which are run offline on randomly sampled models. The output trajectories are then recorded in a dataset accessible from a separate, “voting” website. When users visit the voting website, they are presented with a randomly sampled task and its two associated trajectories, allowing them to vote for their preferred trajectory. This allows us to collect multiple votes for each trajectory. This strategy may require significantly less user effort, but requires users to evaluate tasks submitted by others. In this approach, since users vote based on pre-generated GIFs that are only a few seconds in length, we believe the runtime may be faster than existing Arena platforms, which require generating user prompts on the fly. This approach requires significantly less user interaction time, though users may have less incentive to evaluate tasks provided by others.
>
> We believe an actual implementation of this benchmark may involve a combination of both strategies – users may submit tasks and view the outcomes themselves, but we could subsequently have a much larger pool of users evaluate these same tasks to improve sample efficiency.
>
> **Regarding Multimodal Observations**
>
> Currently, all users are provided with GIFs and agent trajectories for all executed tasks, allowing them to evaluate models based on their interactions with different websites. The models (with the exception of DeepSeek-R1), are provided with screenshots of the webpages they visit, alongside the contents of the webpage. In our opinion, the platform supports multimodal input both for models acting on the open web and the users voting on model preferences - we would be grateful if you could clarify which limitations you believe occur due to a lack of multimodal observation and grounding.
>
> **Regarding the Use of the Browser Use Framework**
>
> For our paper, we chose an open-source framework for transparency in the code, and Browser Use is one of the primary open-source frameworks available for orchestrating agents. As the benchmark scales, we are happy to integrate other agent frameworks and maintain leaderboards in each setting.
> However, we believe that our main contribution in this paper is not necessarily the tooling around the benchmark, but the proposition that there are fundamental differences in how different LLMs behave when performing tasks as web agents, and we can surface these differences using our approach, independent of the tooling used. As a result, while some behaviors or failure modes that were surfaced may be mitigated by using a specific agent framework, we still think that our approach surfaces how language models differ in their behavior under frequently occurring scenarios.

---

> > ### Author Response · Authors · 2025-11-21
> > **Response (2/3)**
> >
> > **Regarding the Validity of Step-Level Feedback**
> >
> > We would like to clarify that we never measure the entire success of a user-provided task based on a single step. The step-level success that we ask users to label is simply whether the LLM achieved the LLM-generated goal for that step. The LLM-generated goal is often a very immediate one (such as navigating back to the search results or performing a search with a different query) whose success is often easy for a human to evaluate qualitatively. In the event that the model initially fails to achieve the goal in the first step and recovers with a similar goal in the next step, we expect users to mark the first step as incorrect and the next as correct. In this way, we are also able to identify when models recover or improve upon their performance in the initial steps by combining patterns in the user's step-level feedback with their votes.
> >
> > Task-level success is strictly measured in terms of the user's vote between the two competing agents, and it is this task-level success that is used to construct the leaderboard. As a result, if there are LLMs skilled in recovery, we expect them to exhibit patterns of “incorrect, correct” occurrences repeatedly in their trajectories, and for users to consistently prefer them over other models. We believe that these should be discoverable from our existing data.
> >
> > **Regarding Distributional Insights**
> >
> > We provide additional distributional insights into LLM agent behavior, as well as task diversity and difficulty. First, we cluster the actions taken by different agents in their trajectories using Docent. A complete description of the available browser actions is presented in Table 1. We observe that a plurality of actions taken involve information collection and form interaction, with other browser actions occurring less frequently.
> >
> > $$
> > \\begin{array} {|r|r|}\\hline \\text{Browser Action Type} & \\text{Percentage of total actions} \\\\ \\hline \\text{Search-Based Information Gathering (search\\_google, click\\_element\\_by\\_index, extract\\_page\\_content)} & 24.3\\% \\\\ \\hline \\text{Direct Website Navigation (go\\_to\\_url, extract\\_page\\_content)} & 12.3\\% \\\\ \\hline \\text{Form Interaction and Input (input\\_text, click\\_element\\_by\\_index, click\\_element\\_by\\_xpath, click\\_element\\_by\\_selector)} & 20.7\\% \\\\ \\hline \\text{Multi-Tab Browsing (switch\\_tab, close\\_tab, open\\_url\\_in\\_new\\_tab)} & 3.6\\% \\\\ \\hline \\text{Page Navigation and Scrolling (scroll\\_down, scroll\\_up, go\\_back, scroll\\_to\\_text)} & 7.2\\% \\\\ \\hline \\text{Advanced Browser Controls (wait, wait\\_for\\_element\\_to\\_be\\_visible, send\\_special\\_keys, get\\_dropdown\\_options, select\\_dropdown\\_option\\_by\\_text, drag\\_and\\_drop)} & 1.1\\% \\\\ \\hline \\text{Misc (e.g. Content Preservation, Task Completion)} & 30.7\\% \\\\ \\hline \\end{array}
> > $$
> >
> > Second, we classify the tasks based on the mean number of steps taken by the two agents assigned to the task. Given that we limit the maximum number of steps taken by an agent to 15, we categorize user-submitted tasks as simple, medium, and complex based on whether the mean number of steps taken by two agents on the task lies in the ranges 0-5, 5-10, or 10-15.
> > Percentage of tasks in each category: Simple: 42.20%, Medium: 43.12%, Complex: 14.68%.
> >
> > Sometimes, however, tasks can fail fast within a small number of steps, which may obscure the true complexity of the task. To analyze task complexity, we also present a second method: measuring the mean percentage of steps an agent executes “correctly” while attempting to solve the task. We refer to this metric as the mean correctness percentage, where correctness is evaluated based on user-provided annotations. We classify tasks into three classes based on the mean correctness percentage: Easy (>66.66% mean correctness), Medium (33.33-66.66% mean correctness) and Hard (<=33.33% mean correctness). Percentage of tasks in each category: Easy: 49.54%, Medium: 42.20%, Hard: 8.26%.

---

> ### Author Response · Authors · 2025-11-21
> **Response (3/3)**
>
> **Topic Distribution of User-Submitted Tasks**
>
> We also cluster the user-submitted tasks using Docent and categorize them into the following types of tasks. The most prominent categories include list/ranking compilations, educational content and summaries, and news headlines and current events. These topics may be potential sources of biases in our dataset.
>
> $$
> \\begin{array} {|r|r|}\\hline \\text{Cluster} & \\text{Percentage} \\\\ \\hline \\text{News headlines and current events} & 10.5\\% \\\\ \\hline \\text{Product price comparison across retailers} & 2.9\\% \\\\ \\hline \\text{Flight and travel booking comparison} & 6.6\\% \\\\ \\hline \\text{Top-ranked lists and rankings} & 32.5\\% \\\\ \\hline \\text{Sports statistics and player information} & 5.9\\% \\\\ \\hline \\text{Entertainment content discovery} & 8.8\\% \\\\ \\hline \\text{Product specifications and reviews} & 4.9\\% \\\\ \\hline \\text{Recipe and cooking information} & 1.5\\% \\\\ \\hline \\text{Real estate and accommodation listings} & 2.7\\% \\\\ \\hline \\text{Weather and location-specific information} & 4.6\\% \\\\ \\hline \\text{Financial and market data} & 1.5\\% \\\\ \\hline \\text{Educational content and summaries} & 11.0\\% \\\\ \\hline \\text{Event listings and schedules} & 2.2\\% \\\\ \\hline \\text{Transportation schedules and routes} & 4.4\\% \\\\ \\hline \\end{array}
> $$

---

### Official Review · Reviewer_igzk · 2025-11-03

**Soundness:** 4
**Presentation:** 4
**Contribution:** 3
**Rating:** 6
**Confidence:** 3

**Summary:**

This paper presents an evaluation platform, BrowserArena, that collects user preference data on 109 user-submitted tasks to construct a language model leaderboard.
They proposed a new method for evaluating LLM performance in web browsing by collecting step-level user annotations on agent traces and analyzing them to identify failure modes.

**Strengths:**

- The dataset is well-motivated and moves toward challenges that are better representative of real-world tasks instead of sandbox tasks.
- This work focuses on failures in CAPTCHA, pop-ups, and navigation, which are overlooked in many other benchmarks.

**Weaknesses:**

- Adding more information on data quality and distribution would be helpful.

- Authors need to add more details and information to ensure reproducibility.

**Questions:**

- What happens if both responses from agents are bad?

- Most of the figures are not readable and need larger font/better color selection.

**Details Of Ethics Concerns:**

Live browsing and solving captchas raises ToS, privacy, and safety and legal issues, which have not been discussed properly in the paper.

---

> ### Author Response · Authors · 2025-11-21
> **Response (1/3)**
>
> Thank you for your feedback and for recognizing that we surface issues overlooked by other benchmarks.
>
> **If both agent responses are bad**
>
> Users have the option to vote for Left, Right, or Tie based on the agent trajectory. The original LMArena platform has a button for “Both are bad”, but we ask users not to vote for that option, since our goal is to identify both partial task completion and completely correct task execution. We ask users to vote for whichever model came closest to task completion. If both agents are equally bad, then we expect users to vote for a Tie.
>
> **Reproducibility**
>
> We commit to updating the paper with our complete Qualtrics questionnaire and the annotation interface. Our post-processing pipeline for failure mode discovery is described in Appendix I. We do not modify the responses received on Qualtrics except to correct any formatting issues with the step-level feedback and complete any missing or mislabeled task IDs and prompts in the user submission using the log files that recorded the task on our evaluation system.
>
> **Figures**
>
> We will improve the quality and resolution of the provided figures.
>
> **Analysis of Task Difficulty and LLM Trajectory Distributions**
>
> We provide additional distribution information for LLM agent behavior, including task diversity and difficulty. First, we cluster the actions taken by different agents in their trajectories using Docent. A complete description of the available browser actions is presented in Table 1. We observe that a plurality of actions taken involve information collection and form interaction, with other browser actions occurring less frequently.
>
> $$
> \\begin{array} {|r|r|}\\hline \\text{Browser Action Type} & \\text{Percentage of total actions} \\\\ \\hline \\text{Search-Based Information Gathering (search\\_google, click\\_element\\_by\\_index, extract\\_page\\_content)} & 24.3\\% \\\\ \\hline \\text{Direct Website Navigation (go\\_to\\_url, extract\\_page\\_content)} & 12.3\\% \\\\ \\hline \\text{Form Interaction and Input (input\\_text, click\\_element\\_by\\_index, click\\_element\\_by\\_xpath, click\\_element\\_by\\_selector)} & 20.7\\% \\\\ \\hline \\text{Multi-Tab Browsing (switch\\_tab, close\\_tab, open\\_url\\_in\\_new\\_tab)} & 3.6\\% \\\\ \\hline \\text{Page Navigation and Scrolling (scroll\\_down, scroll\\_up, go\\_back, scroll\\_to\\_text)} & 7.2\\% \\\\ \\hline \\text{Advanced Browser Controls (wait, wait\\_for\\_element\\_to\\_be\\_visible, send\\_special\\_keys, get\\_dropdown\\_options, select\\_dropdown\\_option\\_by\\_text, drag\\_and\\_drop)} & 1.1\\% \\\\ \\hline \\text{Misc (e.g. Content Preservation, Task Completion)} & 30.7\\% \\\\ \\hline  \\end{array}
> $$
>
> Second, we cluster the tasks based on the mean number of steps taken by the two agents assigned to the task. Given that we limit the maximum number of steps taken by an agent to 15, we categorize user-submitted tasks as simple, medium, and complex based on whether the mean number of steps taken by two agents on the task lies in the ranges 0-5, 5-10, or 10-15.
> Percentage of tasks in each category: Simple: 42.20%, Medium: 43.12%, Complex: 14.68%.
>
> Sometimes, however, tasks can fail fast within a small number of steps, which may obscure the true complexity of the task. To analyze task complexity, we also present a second method: measuring the mean percentage of steps an agent executes “correctly” while attempting to solve the task. We refer to this metric as the mean correctness percentage, where correctness is evaluated based on user-provided annotations. We classify tasks into three classes based on the mean correctness percentage: Easy (>66.66% mean correctness), Medium (33.33-66.66% mean correctness) and Hard (<=33.33% mean correctness). Percentage of tasks in each category: Easy: 49.54%, Medium: 42.20%, Hard: 8.26%.

---

> > ### Author Response · Authors · 2025-11-21
> > **Response (2/3)**
> >
> > **Analysis of Task Topics**
> >
> > We also cluster the user-submitted tasks using Docent and categorize them into the following types of tasks. The most prominent categories include list/ranking compilations, educational content and summaries, and news headlines and current events.
> >
> > $$
> > \\begin{array} {|r|r|}\\hline \\text{Cluster} & \\text{Percentage} \\\\ \\hline \\text{News headlines and current events} & 10.5\\% \\\\ \\hline \\text{Product price comparison across retailers} & 2.9\\% \\\\ \\hline \\text{Flight and travel booking comparison} & 6.6\\% \\\\ \\hline \\text{Top-ranked lists and rankings} & 32.5\\% \\\\ \\hline \\text{Sports statistics and player information} & 5.9\\% \\\\ \\hline \\text{Entertainment content discovery} & 8.8\\% \\\\ \\hline \\text{Product specifications and reviews} & 4.9\\% \\\\ \\hline \\text{Recipe and cooking information} & 1.5\\% \\\\ \\hline \\text{Real estate and accommodation listings} & 2.7\\% \\\\ \\hline \\text{Weather and location-specific information} & 4.6\\% \\\\ \\hline \\text{Financial and market data} & 1.5\\% \\\\ \\hline \\text{Educational content and summaries} & 11.0\\% \\\\ \\hline \\text{Event listings and schedules} & 2.2\\% \\\\ \\hline \\text{Transportation schedules and routes} & 4.4\\% \\\\ \\hline  \\end{array}
> > $$

---

> > > ### Author Response · Authors · 2025-11-21
> > > **Response (3/3)**
> > >
> > > # Discussion of Ethical Concerns
> > >
> > > We first discuss ethical concerns in the context of data that has already been collected for this paper.
> > >
> > > **Privacy Concerns**
> > >
> > > For the data collected via the paper, the platform explicitly asks users not to enter personal or sensitive information, such as logins and passwords, and does not provide an interface for users to log in to specific websites. We carefully monitored responses during our study to ensure these rules were followed and detected no violations. As a result, none of our data includes any personal information about participants or other private individuals; all information collected is publicly accessible. We also monitored to ensure that user-submitted tasks did not modify any state (e.g., posting messages/comments online); again, there were no violations.
> > >
> > > **Terms of Service Concerns**
> > >
> > > For the paper, the data collected for the paper was limited by restricting the number of users who could access the platform at a given point in time, ensuring that it was not possible for any single website to be flooded with multiple requests from our platform. Additionally, since we do not enable logins for any services, we don’t believe that any of the trajectories generated for our tasks raise Terms of Service concerns.
> > >
> > > ## Scaling the BrowserArena Platform
> > >
> > > In the following discussion about some of the ethical issues that may occur when we scale BrowserArena, we propose two approaches to scaling our benchmark. The first, referred to as the “one-step approach”, is the format used for collecting data in the paper, where users submit tasks to two randomly selected LLMs and vote for their preferred output. During the data collection process for our paper, we limited the number of users on our platform at any given time and manually ensured compliance with ethical principles. In our discussion, we describe ways to automate this compliance when we scale to levels where it is no longer feasible to manually monitor data collection efforts.
> > >
> > > The second approach, which we shall refer to as the “two-step” approach, involves collecting tasks from users separately from the votes. Users submit tasks, which are run offline on randomly sampled models. The output trajectories are then recorded in a dataset accessible from a separate, “voting” website. When users visit the voting website, they are presented with a randomly sampled task and its two associated trajectories, allowing them to vote for their preferred trajectory. This allows us to collect multiple votes for each trajectory, and also implement more extensive filters and safeguards for tasks.
> > >
> > > **Privacy**
> > >
> > > To scale using the one-step approach, we plan to establish an LLM-based filter to remove any task prompts that violate these conditions, similar to how LMArena filters out prompts considered harmful. To scale using the two-step approach, we will also extensively search over trajectories to ensure that no websites visited or discovered during the agent’s search expose any private information. This will be achieved by having multimodal models flag such task trajectories using the trajectory text and GIF output.
> > >
> > > **Terms of Service**
> > >
> > > If we were to scale using the one-step approach, we acknowledge that it would be difficult to comply with ToS terms that forbid any form of web scraping. In this setting, we would remove the browser's ability to directly navigate to a website and instead only enable it to perform Google Searches that respect the website's robots.txt setting. Additionally, we would terminate tasks on any detection of CAPTCHAs or bot protection in the trajectory. In the two-step approach, we would include additional checks on lists of websites to be visited and filter out any tasks that may require visiting websites that do not allow scraping.

---

> > > > ### Author Response · Authors · 2025-12-03
> > > > **Qualtrics Questions**
> > > >
> > > > The questions used to collect data in our Qualtrics form are available in the updated version of the paper in Appendix P. The agent action clustering results and the task clustering results have been added to Appendix N and O respectively.

---

### Meta-Review · Area_Chair_kpwY · 2026-01-15

**Summary:**

- Reviewer igzk raised a question regarding the benchmark handling of "both bad" comparisons, for which the authors responded that the intention is give partial credits upon failures. I share a more fundamental concerns with the evaluation design: arena-style metrics, such as chatbot-arena, are already under a certain level of controversy of being subject to human preferential biases that is not fully correlated to the success of tasks, thus undermining the value of derived ELO scores. In the paper, the authors also indicates that a satisfying inter-annotator agreement is only achieved when "tied" judgements are removed, which also indicates that there is a fair amount of variance (because when tied is annotated, a "forced judgement" might goes to either sides). This is a major concern I share.
- Reviewer G5Uu raised a critical concern that the failure modes discovered in this paper such as captcha and pop-up closure, are not central to the web agent capability
- Reviewer jXW9 raised a concern that the scaffolding (Browser Use) undermines the validity of the evaluation, because it introduces an added confounding factor of model's capability to use the framework in addition to acting as a web agent. This is a valid concern that also exists in other domains such as SWEBench, but in practice SWEBench does not force a scaffolding. This is another impactful concern.

**Reviewer Concerns:**

Reviewer igzk's two weakness points and second question was addressed. the first question, as I listed in the summary, remains a valid concern.

Among Reviewer jXW9's three weakness points. The first point is insightful, but the weakness argument itself (users have less motivation to compare web agents as opposed to a general chatbot agent) is somewhat less supported by clear evidences. Thus I do not consider this as a major weakness of the paper. The seond point is indeed a little vague as the authors pointed. The third point was not addressed and remains a concern. two questions are addressed by the rebuttal.

For Reviewer G5Uu's comments, the first point is a critical concern not addressed. The second point is valid, but is perpendicular to this work's main contribution (a live benchmark for web agent instead of improved metrics for evaluation). However, it does reflect the shared concern in the community about the arena-style benchmarks, showing the limitation of this work. The third point should not be a major concern. Although scale is important, requiring a large scale for the first version of a benchmark might limit acceptable work to only coming from large companies and has the risk of missing those small scale work with valuable insights.

**Reviewer Scores:**

I think the reviewers are going to keep their scores.

---

### Decision · Program_Chairs · 2026-01-26

Reject